# Combining natural language processing and metabarcoding to reveal pathogen-environment associations

**David C. Molik**[1,2]*, **DeAndre Tomlinson**[3☯], **Shane Davitt**[1☯¤], **Eric L. Morgan**[2], **Matthew Sisk**[2], **Benjamin Roche**[4], **Natalie Meyers**[2], **Michael E. Pfrender**[1]

**1** Department of Biological Sciences, University of Notre Dame, Notre Dame, Indiana, United States of America, **2** Navari Center for Digital Scholarship, University of Notre Dame, Notre Dame, Indiana, United States of America, **3** Science-Computing Program, University of Notre Dame, Notre Dame, Indiana, United States of America, **4** Cold Spring Harbor Laboratory, Cold Spring Harbor, New York, United States of America

☯ These authors contributed equally to this work.
¤ Current address: School of Law, University of Texas at Austin, Austin, Texas, United States of America
* dmolik@nd.edu

**Data Availability Statement:** The data underlying the results presented in the study are available from osf.io at doi.org/10.17605/OSF.IO/29V3F and doi.org/10.17605/OSF.IO/49W5R.

## Abstract

*Cryptococcus neoformans* is responsible for life-threatening infections that primarily affect immunocompromised individuals and has an estimated worldwide burden of 220,000 new cases each year—with 180,000 resulting deaths—mostly in sub-Saharan Africa. Surprisingly, little is known about the ecological niches occupied by *C. neoformans* in nature. To expand our understanding of the distribution and ecological associations of this pathogen we implement a Natural Language Processing approach to better describe the niche of *C. neoformans*. We use a Latent Dirichlet Allocation model to *de novo* topic model sets of metagenetic research articles written about varied subjects which either explicitly mention, inadvertently find, or fail to find *C. neoformans*. These articles are all linked to NCBI Sequence Read Archive datasets of 18S ribosomal RNA and/or Internal Transcribed Spacer gene-regions. The number of topics was determined based on the model coherence score, and articles were assigned to the created topics via a Machine Learning approach with a Random Forest algorithm. Our analysis provides support for a previously suggested linkage between *C. neoformans* and soils associated with decomposing wood. Our approach, using a search of single-locus metagenetic data, gathering papers connected to the datasets, *de novo* determination of topics, the number of topics, and assignment of articles to the topics, illustrates how such an analysis pipeline can harness large-scale datasets that are published/available but not necessarily fully analyzed, or whose metadata is not harmonized with other studies. Our approach can be applied to a variety of systems to assert potential evidence of environmental associations.

**Funding:** The author(s) received no specific funding for this work.

**Competing interests:** The authors have declared that no competing interests exist.

## Author summary

We expand the utility of Natural Language Processing (NLP), backtracking through meta-barcodes, utilizing papers that may not mention our subject of interest, *C. neoformans*, in a departure from usual text analysis methods. We confirm that *C. neoformans* is associated with decomposing wood which is reinforced by the inferred literature studied here on *C. neoformans* and its close congeneric relatives. This work demonstrates the potential utility of pairing NLP with single-locus metagenetic data for the study of Neglected Tropical Diseases. While the results of this article are largely confirmatory, we present a novel method to study the ecological niches of rare pathogens that leverages the immense amount of data available to researchers in the NCBI Sequence Read Archive (SRA) combined with a text-mining analysis based on Natural Language Processing. We demonstrate that text processing, noun identification, and verb identification can play an important role in analyzing a large corpus of documents together with metagenetic data. Forging this connection requires access to all of the available ecological 18S ribosomal RNA and Internal Transcribed Spacer NCBI SRA datasets. These datasets use metabarcoding to query taxonomic diversity in eukaryotic organisms, and in the case of the Internal Transcribed Spacer, they specifically target Fungi. The presence of specific species is inferred when diagnostic 18S or ITS gene region sequences are found in the SRA data. We searched for *C. neoformans* in all 18S and ITS datasets available and gathered all associated journal articles that either cite the SRA data accessions or are cited in the SRA data accessions. Published metagenetic data often have associated metadata including: latitude and longitude, temperature, and other physical characteristics describing the conditions in which the metagenetic sample was collected. These metadata are not always presented in consistent formats, so harmonizing study methods may be needed to appropriately compare metagenetic data as commonly required in metanalysis studies. We present an analysis which takes as input articles associated with SRA datasets that were found to contain evidence of *C. neoformans*. We apply NLP methods to this corpus of articles to describe the niche of *C. neoformans*. Our results reinforce the current understanding of *C. neoformans*'s niche, indicating the pertinence of employing an NLP analysis to identify the niche of an organism. This approach could further the description of virtually any other organism that routinely appears in metagenetic surveys, especially pathogens, whose ecological niches are unknown or poorly understood.

## Introduction

From the inception of the study of public health and epidemiology two major challenges have been of paramount importance: identifying the origin of the pathogen and identifying the determinants of transmission. This work focuses on origin. An early example of confronting and overcoming these challenges through evidence and inference can be found classically in Dr. John Snow's observations of cholera in 19th century England. A major contributor to the foundations of modern epidemiology, Snow is perhaps best known for his role in fighting the infamous 1854 Broad Street cholera outbreak; Confronted with a devastating outbreak of cholera, Snow sought to determine the origin of the outbreak. Drawing on a body of indirect evidence he hypothesized that cholera was transmitted via contaminated water [1]. His hypothesis was based on statistical evidence linking water supply companies and water sources to an increase in deaths [2]. In the modern era, indirect evidence again played an important role in establishing the link between El Niño and cholera outbreaks in South America in the

1990s, informing optimal strategies for vaccination [3,4]. Similarly, we are informed by this tradition of non-obvious inference as we pursue the niche of *Cryptococcus neoformans* through the use of Natural Language Processing (NLP).

*Cryptococcus neoformans*, a basidiomycete dimorphic yeast (subphylum Agaricomycotina), was first described by Francesco Sanfelice [5], and transferred to genus *Cryptococcus* by Jean Paul Vuillemin in 1901 [6]. It is the principal causative agent of cryptococcosis [7], a usually coinfecting disease of meningitis in immunocompromised individuals, principally in HIV/ AIDS patients [8]. This pathogen is responsible for life-threatening infections and has an estimated worldwide burden of 220,000 new cases each year, with 180,000 resulting deaths. The African Continent is home to 162,500 new Cryptococcosis involved cases a year [9] with the majority of worldwide deaths occurring in sub-Saharan Africa [9]. For such a prevalent and destructive pathogen, it is somewhat surprising that the life-cycle in nature, and ecological niches occupied by *Cryptococcus spp.* are not yet fully understood [10]. *C. neoformans*'s "classic" habitat is thought to be soil and avian guano [10]. However, the closely related species *Cryptococcus gattii* [11], responsible for the Vancouver Island (BC, Canada) outbreak of cryptococcosis in 1999, is documented to be associated with the bark of a variety of tree species (potentially over 50) [12]. Several recent studies have also found *C. neoformans* isolates in association with trees, such as eucalyptus in Egypt [13], and olive in Turkey [14]. *(note: at least a dozen such papers)* Furthermore, recent work has shown that *C. neoformans* is able to grow both on live plant material, such as *Arabidopsis* seedlings and Douglas fir trees, as well as in saprobic association with dead plant materials [15]. Therefore, the ecological niche of *Cryptococcus* may be broader than previously recognized. While it is posited that most cases of cryptococcosis are not typically associated with a specific known environmental exposure [10], the observance of specific ecological associations could be obscured because Cryptococcosis typically affects immunocompromised individuals, so sampling bias could be introduced.

Previous work shows a community-level linkage between *C. neoformans* and woody decomposers [16–18]. While *C. neoformans* is generally associated with tree bark and soil, our results suggest that these are possibly common environments for *C. neoformans*, and that consequently the range of *C. neoformans* may be larger than expected, as evidenced by aerosol sampling in Columbia [19]. *C. neoformans* is hypothesized to be an accidental pathogen. The accumulation of genomic traits that make it an effective pathogen, for example, traits such as the ability to form capsules and cross the Blood Brain Barrier via infected phagocytes [20–22], may have been acquired during its evolutionary history as a result of selective pressures unrelated to pathogenicity in human hosts, A greater understanding of the features of *C. neoformans*'s natural habitat will aid research into the accidental pathogenesis hypothesis [20,21,23]; This connection between virulence and the environment necessitates a deeper understanding of the particular environmental associations of the pathogen.

The distribution, number of species, and phylogenetic relationships among members of the *Cryptococcus* species complex have been difficult to accurately define, with recent propositions that *C. neoformans* and *C. gattii* are themselves polyphyletic aggregates representing at least seven different species [24]. Prior to the Vancouver Island outbreak, *C. gattii* had only been reported from tropical and subtropical regions [25]. This gap in observation shows that not only do cryptic environmental niches exist, but they could also have serious consequences for the epidemiology of cryptococcosis, lending weight to the argument that *C. neoformans* may have a broader range of habitats than is currently recognized. The classical method to identify *Cryptococcus* in nature has been the recovery and culture of natural isolates. The recent expansion in wealth and breadth of environmental metagenetic datasets now makes it possible to further uncover some of these cryptic niches, especially from studies not primarily targeted towards *Cryptococcus* spp. In this study we use NLP to more easily overcome some of the

hurdles of a more traditional meta-analysis. Published sample data often lacks extensive descriptions of machine-readable metadata, sampling methods, or the physical measurements (e.g. salinity, temperature). Knowing this truism, we "look in" or computationally process descriptions of metagenetic samples referred within the main body of articles. NLP can circumvent differences in scientific units and article format among studies that can cause errors in traditional meta-analysis. An extreme example of such a situation would be a meta-analysis based on articles which recorded temperature in either degrees Kelvin or Celsius. Both are agreed scientific standards, and without metadata it is possible to mistake one for the other, even computationally. By using NLP on the body of the text of the article we can ignore such conflicts in the metadata. In our analysis, we make an assumption that the text in journal descriptions has a connection to the presence of *C. neoformans* in the metagenetic data used for the study. The assertion of this link could be misleading if the paper associated with the dataset did not actually describe the conditions in which the samples were found, if say, a dataset was associated after the publishing of the article: an important factor in the lack of metadata and description of the sample conditions.

In this study, the *C. neoformans*–woody decomposition association is expounded via a machine learning text-categorization analysis. To gather the data (the text of journal articles) for this analysis, the 18S portion of the eukaryotic ribosomal subunit (18S) and the Internal Transcribed Spacer (ITS, although note that in most papers, this one included, ITS usually refers to the 2nd ITS sometimes denoted as ITS2) of the reference sequences of *C. neoformans* were used as query sequences for the entirety of National Center for Biotechnology Information (NCBI) Sequence Read Archive (SRA) [27] collection of metagenetic single-locus datasets of either 18S or ITS amplicons. The papers associated with the datasets containing *C. neoformans* DNA sequences, as indicated by the SRA search, were then collected and utilized in a random forest analysis. This analysis revealed that studies which found *C. neoformans* can be associated by their shared mentioning of terms indicating woody decomposition. This method, utilizing papers associated with metagenetic datasets, is useful in part because it reveals studies that found *C. neoformans* that might not explicitly mention *C. neoformans*. In our case, notably, only one paper mentioned *C. neoformans* within the body of its text necessitating an analytic approach that goes beyond simple search of papers that mention *C. neoformans*. In this study, the ecology of *C. neoformans* was elucidated using data from metabarcoding studies (for more information on metabarcoding see Box 1). Such an approach allows us to garner insights into species that are biomedically important and not well-studied ecologically.

### Box 1. "What is Metabarcoding?"

Metagenetic single-locus data generally falls under the method of Metabarcoding. Metabarcoding is a method of determining the taxonomic diversity within the contents of a sample by the analysis of the DNA sequence from a specific gene region [26]. This is much like a primer that might be used to find a specific species, but instead a generalized primer is used. Searching for Fungi with metabarcoding is usually done through two common gene regions, the 18S and the ITS; both are portions of the Eukaryotic ribosomal-associated DNA in all Eukaryotes. The technique of metabarcoding is mostly used in the analysis of microbiomes both for prokaryotes, archaea as well as eukaryotes. Metabarcoding has been used to locate multicellular organisms as well [26]. The datasets that were combed through for the presence of *C. neoformans* are the eukaryotic fractions of microbial communities.

## Methods

We use NLP to determine "topics" that are associated with positive labeled papers, or papers associated with SRA samples that contained *C. neoformans*, and Random Forests (made of decision trees, for more information on decision trees see Box 2), to assess the quality of these

### Box 2. What is a Decision Tree?

Decision trees operate on an entropy-reduction basis or a probability maximization basis, depending on the type. Decision trees split datasets or information by calculating which feature lowers the entropy of the system the best information, until the entropy is at a minimum [29]. In ID3 decision trees, information gain is the change in entropy from a previous state to a new state based on a condition, like a feature in a dataset [29]. Information gain determines the starting feature to be split by comparing the entropies between all the classes, and choosing the split based on the feature that provides the lowest entropy, which is the highest information gain. CART (Classification and Regression Trees) use a similar method called a Gini index, which separates nodes based on subtracting the sum of the squared probabilities from each class by 1 [43]. The feature with the lowest gini index is chosen for a split, and this continues until the gini index equals 0. The problem with both decision tree approaches is the dependence on the initial starting feature to be split. By the nature of the calculation, it is susceptible to overfitting due to class imbalances and the initial starting node. Random forest resolves these two disadvantages by using a bootstraps aggregation (i.e. bagging) method [44]. It creates multiple decision trees with different starting nodes, and uses a majority vote system for the final prediction. This approach resolves both issues because it accounts for different starting points, and by having multiple randomly generated models running concurrently, overfitting is reduced even with class imbalances. The overfitting that one tree would have in the forest is mitigated by a different tree that did not overfit; this method is similar to taking an average of many different trees all at once, but more robust.

"topics" (see Fig 1, for visualization of analysis steps). Decision Tree NLP techniques read and pseudo-comprehend human text and language to derive meaning and discover interactions through machine learning. Two areas that NLP focuses on are word syntax and word semantics [28]. The syntax of a sentence refers to the arrangement of words. Semantics refers to the meaning derived from the words and their arrangement. Model creation algorithms such as Latent Dirichlet Allocation (LDA) can be used as classifiers and predictors [28]. We use LDA in the *de novo* determination of the number of "topics" (henceforth: topics) or rough association of journal articles within our corpus of documents. We then utilize random forest in our machine learning text-categorization pipeline to assign journal articles to the LDA determined topics. The topics which have the bulk of papers containing the unstated connection to *C. neoformans* are the topics from which environmental niche associated words are drawn. Decision Tree Learning is a reliable evaluation method for classification methods and assessing model performance. Decision trees have been powerful classifiers in machine learning algorithms since 1975 [29]. Ensemble methods, a composite of several methods, have improved the performance of individual models by different means, notably by bootstraps [30]. Random forest is an adaptation of decision trees with a bootstraps advantage. Bootstraps involve creating multiple decision trees from the sample data and using majority voting to determine the best split

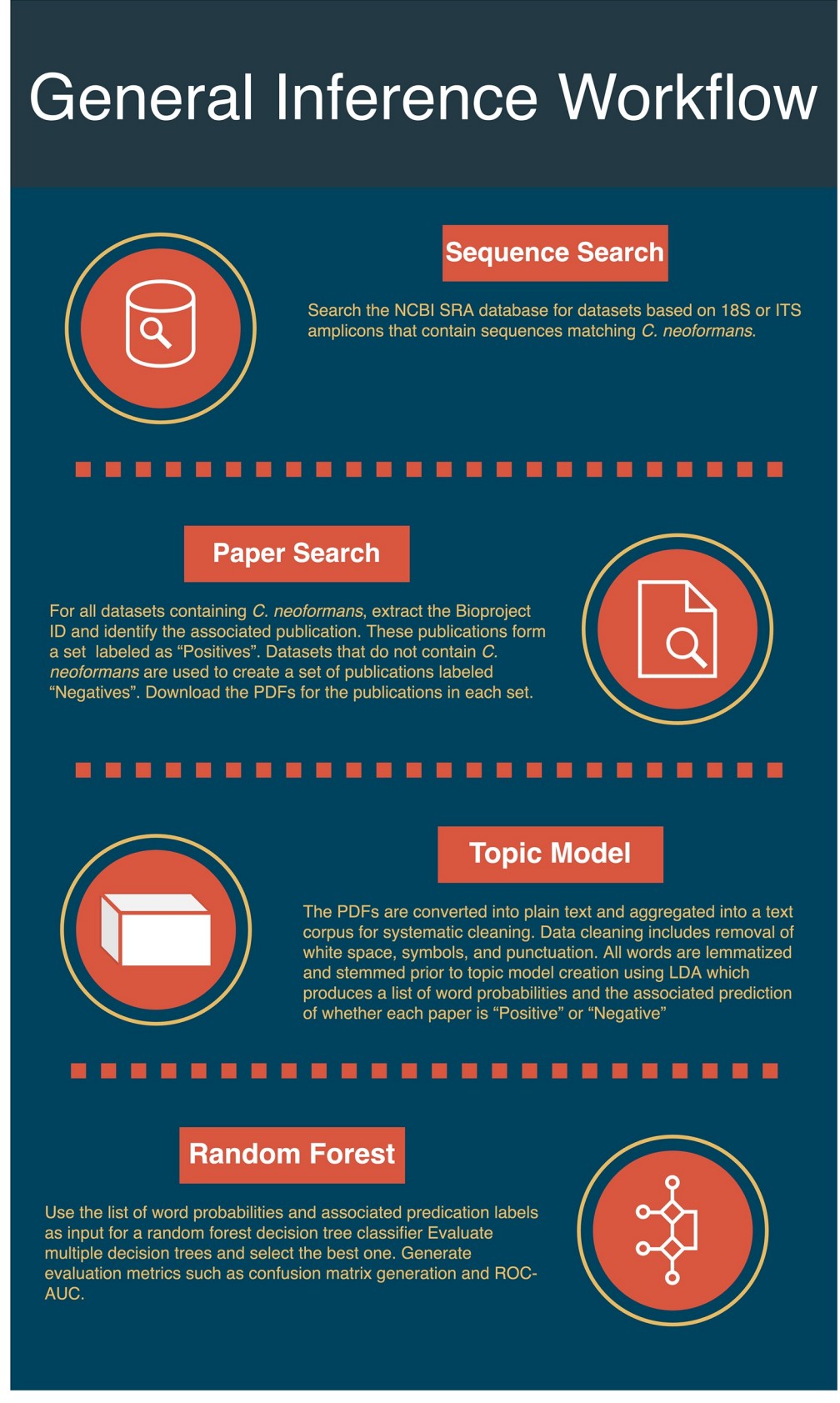

**Fig 1. General Workflow for associating articles into topics, and then determining topic and paper assignment validity.** Four overarching actions are taken in this analysis: the initial barcode search, the results of which are used in a paper search, the resulting papers of which are used in a topic model, and finally the topic model is validated with a random forest model.

at each node in the tree [30]. With the ability to create hundreds of trees from varied samples of the training data, the overall performance of the final model is less over-fitted and has better performance than a single decision tree. In a departure from pure NLP research we determined the number of topic clusters via NLP evaluation and confirmatory visualization methods, and found that the majority of our positive hit papers formed a single topic. The results indicated that there was a combination of words that drew these papers together. Furthermore, these articles could be recalled and predicted reliably, indicating a measure of validity.

## Searching NCBI SRA, collecting articles

Our initial task was to collect datasets that contain *C. neoformans*. We created a script that crawled the NCBI SRA for datasets generated using universal 18S and fungal specific ITS PCR primers. This search was denoted by the query run on the NCBI SRA database in October of 2018:

```
"ecological metagenomes" [Organism] AND (18S OR ITS) AND
(cluster public[prop] AND "biomol dna" [Properties])
```

This query was used to find candidate metagenetic datasets, later pared down to just datasets containing *C. neoformans* via BLAST alignment. The SRA identifiers of samples were downloaded in fasta format. Using the command-line utility blastn_vdb, this file of SRA numbers was used to extract the corresponding NCBI datasets and search those datasets against reference sequences of the 18S and ITS regions diagnostic of *C. neoformans*. This search yielded the SRA numbers of the NCBI datasets that were specific (henceforth: positive) for *C. neoformans (see supplement for SRA and gene Ascension numbers)*. The stringency parameters for the searches for both ITS and 18S were a percent identity value of 98% and an e-value of 1e-140, more stringent than the commonly used 97% identity for species level identification. A query sequence spanning the entirety of the region was used, which may have limited the number of identified sequences. Of all 18S and ITS possible SRAs at the time of the query 1380 SRA datasets were found to contain a matching sequence to the query sequence.

After retrieving the SRA identifier of datasets that contained *C. neoformans*, the SRA was matched to the Bioproject ID. Many of the 1380 SRA datasets had redundant Bioproject IDs, With the Bioproject ID, the associated published journal article was retrieved via a direct link with the Bioproject ID, or through a search based on a match of date, authors, and sequencing technology published in the article and Bioproject in a google scholar and/or google web search. Articles retrieved via either of these methods were labeled "positive"; only one positive article explicitly mentioned *C. neoformans* in the text of the paper. SRA data that was searched and found to not contain *C. neoformans* was used as "negative." Randomly selected SRA numbers, of all published SRA numbers under "ecological genomes" and "biomol dna", were spot checked for their Bioproject IDs. Since Bioprojects often contain multiple SRA datasets, the other datasets in any given Bioproject had to be checked for the presence of *C. neoformans* diagnostic sequences. These steps ensured negative hits would be as close in format and topic as possible to the positive hits. Papers will typically only reference one Bioproject, but if any of the SRAs within the Bioproject contained *C. neoformans*, that paper would be considered "positive". Random negative SRAs from the list of all SRA were selected, their Bioproject IDs retrieved and the Bioproject-linked papers were retrieved for use as the negative set. PDFs of the positive and negative articles were downloaded. We selected 31 negatively labeled control

papers for inclusion. This step ensured that texts of the negative set would be similar enough to the text of papers in the positive set to enable further analysis driven by properties beyond the basic structure and writing style of the articles in question. Retrieved papers where read for references to sampling locations and environmental types, articles were assigned randomly to two different volunteer analysts for double assessment of location and environment of each journal article thought to be a "positive" hit. The group of analysts was made up of the Authors, as well as members of the Navari Family Center for Digital Scholarship. Since positive hit articles would, by necessity, mention metabarcoding and potentially have a high incidence of description related to metabarcoding processes and environments, there would be a degree of textual similarity. Likewise, as an ostension, comparing the poems of Robert Louis Stevenson [31,32] and the "Positive" hit papers using NLP could be expected to reveal clear differences between the Research Articles and Stevenson's poems driven merely by their time style and time period of content overwhelming the question of whether the poems Robert Louis Stevenson wrote might describe the same differences in niche associated with the presence of *C. neoformans*. In our case the analyzed *C. neoformans* NLP project corpus is all metabarcoding papers, which helps ensure that we can use NLP to examine differences between metabarcoding papers, rather than surfacing subject matter differences between wider ranging research articles.

The associated documents were classified with two binary class labels, "positive" and "negative". We identified a total of 113 papers, but there was a skew towards positive papers resulting in a class imbalance between the positive and negative labeled documents. To rectify this imbalance, of the 113 papers, 82 labeled as Positive hits of *C. neoformans* were used for training the model; 31 labeled as Negative hits were used for testing. Creating the model this way means that we no longer have to account for the class imbalance and instead use the mismatch of our negative hits to measure our accuracy. The downloaded PDF files were converted into text files using an in-house R script with the "TM" library [33]. These raw, preprocessed documents were aggregated into a text corpus using the Gensim python library [34], which allowed for easier manipulation of all the text files at once, while retaining the information and properties of each document. Gensim was chosen as the scientific package of choice for its wide support and ease of use for a variety of machine learning application and use cases, in addition to its memory efficient architecture for multiple trials. To obtain optimal results, data cleaning was performed on the text corpus. The articles were reviewed, and optimized for machine readable grammar (i.e. lemmatized) prior to the aggregation; there are many symbols, characters, whitespace, and words that are not relevant to understanding the essence of a sentence or a topic which were removed [28,35]. Additionally, words were deleted from the corpus if they were noisy, or not relevant to the analysis, such as popular author and publisher names. Afterwards, the corpus was lemmatized and stemmed to root words using Gensim. Lastly, all punctuation and stop words were removed using the NLTK python package [36]. Stop words removed (i.e. not accepted for consideration within the analysis) were the base stop words from the NLTK package.

## Topic modeling

We generated topic models using the Latent Dirichlet Allocation (LDA) method. This algorithm, commonly used in computer science, has additional utility in well-constructed text documents, such as academic journal articles [28]. A Dirichlet distribution is a family of multivariate, continuous, probability distributions used in categorical distributions and Bayesian statistics [40]. It can be generalized as families of continuous probability distributions from "[0, 1]" with multiple variables. For comparison purposes, LDA is specifically designed for

NLP while Principal Component Analysis (PCA) is more general. Consider PCA which is an unsupervised algorithm, like LDA. PCA uses linear transformations to maximize the variance between features and assumes all data objects are continuous. In comparison, LDA uses multivariate continuous distributions and probability distributions, with the assumption that the data objects are discrete and in a "bag-of-words" format.

To determine the topics from the text corpus, LDA essentially reverse engineers the document corpus. The documents to be recreated are represented as a random collection over latent topics, characterized by a distribution over all the words. The documents are represented as a distribution of random words associated with latent topics. When the maximum likelihood that a particular set of random words successfully recreates the document, then those set of words belong to a particular topic.

To summarize (see: Fig 2), **α** and **β** can be compared to matrices, with row **i** and column **j**. In **α**, a value in row **i** and column **j** refers to how likely document **i** contains topic **j**, meanwhile in **β** that value represents how likely topic **i** contains word **j**. The Gensim library has an LDA package which generates the word-topic probabilities, and by mapping the word-topic probabilities to the corresponding documents, the final output of this portion of the method is a list of topic probabilities for the training and testing set with the appropriate "positive" or "negative" label from each document [35]. Gensim was chosen as the scientific package of choice for its wide support and ease of use for a variety of machine learning application and use cases, in addition to its memory efficient architecture for multiple trials.

Since the LDA model in our NLP pipeline is dependent on the number of topics and the optimal number of topics is unknown, it is necessary to create multiple models, each having a different number of topics as an initial parameter. After the model is created, its coherence and perplexity scores are used to evaluate its quality before inputting it into a supervised classifier (for more information on model evaluation see Box 3). Coherence is a measure to give a general understanding of whether the topics are well-defined or unknown. Essentially topic coherence scores show how well the words relate to each other semantically within a topic, and

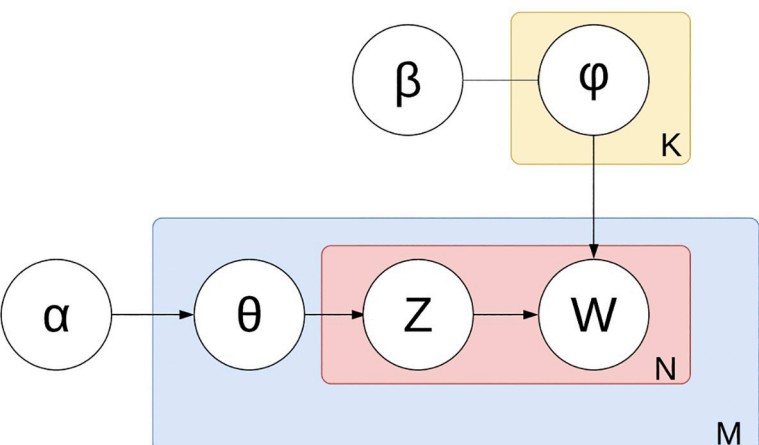

**Fig 2. Plate diagram displaying the conditional probabilistic dependencies in Latent Dirichlet Allocation.** Depicts the core of the LDA method. Plate diagrams are graphical models of probabilistic models with conditional dependence between random variables [37,38]. The boxes are "plates"; they are repeated entities of documents for the outer plate, and words in the inner plate. The Greek symbols denote the hidden or unsupervised areas of LDA, while the regular characters denote the user-created or supervised areas of this algorithm. **α** is the per-document topic distribution [39]. **β** is the per-topic word distribution, and **φ** is the word distribution for a singular topic **K**. Θ is the topic distribution in a particular document **M**, then **Z** is the topic for the **N**-th word in document **M**. These two pathways converge at **W**, which is the specific word. W is the observed variable while every other layer in this process is latent, or unsupervised.

## Box 3. Model Evaluation

Common machine learning practices include model evaluation of the statistical significance of the model classification. Generating a confusion matrix (Table 1) is the starting point that provides a general overview of the model's capabilities. The four possible classifications are true positive (TP), true negative (TN), false positive (FP), and false negative (FN). A true positive indicates the model predicted a data object to be positive and was correct (in our case an article that found *C. neoformans* and was marked as "positive"). A false negative is a positive object that was classified incorrectly as negative value (a "positive" paper classified as negative). Similarly, a false positive is a negative value that was classified as positive (a "negative" paper classified as positive). Lastly, a true negative implies the classifier correctly predicted a negative object correctly (a "negative" paper classified as negative). The truth values in this experiment are the assigned "positive" and "negative" labels from the initial SRA search. The prediction values were binary "positive" and "negative" labels based on the topic probabilities from the LDA model. These groups are tallied, aggregated and analyzed to quantify the performance of the classifier.

It is possible to statistically evaluate the number and proportion of correctly positively identified journal articles from the Random Forest classifier using "Precision" and "Recall." Precision is the proportion of true positive elements. Recall is the true positive rate, which is the proportion of actual positive elements that were correctly identified as positive. These metrics can be skewed when there is a class imbalance, so a third statistic, called the F1 statistic, is the balanced average between precision and recall, and gives an accurate evaluation of how the classifier is performing, since a high F1 scores requires both the precision and recall score be high.

**Table 1. Classic confusion matrix to visually analyze the classification performance of an algorithm.**

|  | Predicted Positive | Predicted Negative |
|---|---|---|
| Ground Truth Positive | True Positive (TP) | False Negative (FN) |
| Ground Truth Negative | False Positive (FP) | True Negative (TN) |

a higher coherence score indicates a better overall model. Model perplexity is a measurement of how well a model can predict unseen data via probability distribution of [41] samples.

## Random forest classification

We used a supervised classification method to quantify how well the LDA model performed. The latent layers of LDA modeling requires an evaluation method to determine the quality of its probability distributions and the created word-topic pairings. Random Forest classification is an adequate method, due to its ability to handle a wide variety of datasets while maintaining preservation of accuracy without overfitting the data. After the LDA topic probabilities were created, they were used as the basis for the random forest model. Since the classes are imbalanced between the "Positive" labeled and "Negative" labeled articles, the training and testing sets maintained a proportional imbalance as well to maintain the integrity of the analysis. The random forest classifier, powered by the Sklearn python library, creates a decision tree model

with the topic probabilities as features and the label as the final positive or negative classification [42].

We created a series of scripts in order to implement LDA topic modeling, run Random Forest Classification, and calculate the AUC-ROC (see supplement). Random Forest is an evolution of the Decision Tree algorithm, in which a number of decision trees are created by randomly selecting portions of the available dataset.

In this experiment, features for the random forest classifier were the topic probabilities, and the label was the binary class label for positive and negative. Since the journal articles are imbalanced between the amounts of "Positive" and "Negative" papers, the training and testing sets maintained a proportional imbalance as well to maintain the integrity of the analysis. The random forest classifier, powered by the sklearn python library with default parameters enabled, classified the LDA model [39,42].

Area Under the Curve-Receiver Operating Characteristic (AUC-ROC) is an important evaluation measurement for binary classification. AUC is the measure of separability, and ROC is a probability curve. When combined, this metric shows how well a model can distinguish between classes. The two components of the AUC-ROC curve are the true positive rate and false positive rate. The true positive rate (TPR) is calculated as the number of true positives divided by the sum of the number of true positives and the number of false negatives. The false positive rate (FPR) is calculated as the number of false positives divided by the sum of the number of false positives and the number of true negatives. The TPR describes how good the model is at predicting the positive class when the actual outcome is positive. The FPR details how often a positive class is predicted when the actual outcome is negative. The ROC algorithm creates the TPR and FPR by using the truth values and the predicted values from the random classifier. Afterwards, the AUC function creates the curvature for the AUC-ROC curve. A classifier with no-skill to determine the difference between a false positive and true positive is linear on the graph, and the higher the skill, the larger the curvature becomes. Another benefit for the AUC-ROC curve is that it is multiple statistics in one. An AUC score is more generalized than the ROC statistics, since the AUC score is similar to the integral of all the curves, while ROC is a visual representation of the true positive and false positive rates at various points in the classification algorithm.

We used a supervised classification method to quantify how well the hidden layers in the LDA model performed. After the LDA topic probabilities were created, this was used as the basis for the random forest model. The Random Forest script assembled multiple decision trees and promoted selection of the optimal one via majority voting for the final classification. Decision trees by themselves have the problem of overfitting training set data with a high variance, but random forest acts as an averaging method to reduce variance [41].

## Results

### Collecting articles

While the metabarcode dataset search resulted in 1380 SRA datasets, each SRA dataset references a Bioproject associated with a paper, and multiple SRA datasets often referenced the same Bioproject ID. The papers came from a wide variety of publishers, with *Elsevier*, *Nature*, *Wiley*, and *Frontiers* contributing more papers than other sources. Two years, 2000 and 2007 seem to be outliers, perhaps caused by authors who assigned new datasets to older papers. The vast majority of studies mentioned that they were based on the 18S genomic region only, rather than ITS solely, both regions, or neither of the regions. The "neither of the regions" category contains papers which are associated with a dataset, where the body of the text does not explicitly mention the association. The soil environment was the primarily discussed

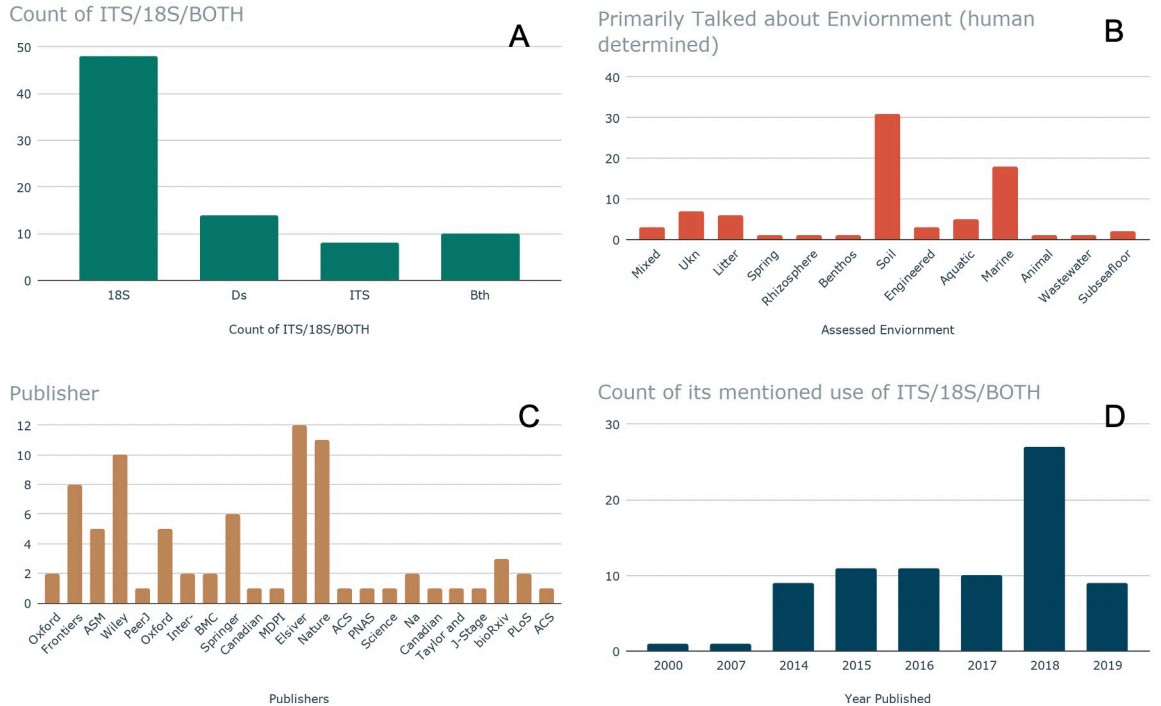

**Fig 3. Examination of positive hit papers.** In panel "A", papers are graphed according to what genomic region, if any, they mentioned in their text, regardless of whether the paper was retrieved via the ITS or 18S region. "Ds" indicates the paper did not mention a genomic region (ie. the paper will have been found via the metabarcoding search and dataset, but the paper may not link back to the dataset itself, this can indicate that the paper was attached to the dataset post publishing). Finally, papers are grouped by what environment they primarily discussed. In Panel "B," A reader assigned the environment, overall, of each environment. Graph in panel "C" indicates the publisher, and Panel "D," the date of publication for the papers used in this study.

environment for a plurality of papers, and the marine environment the second-most-frequent primary environment (see Fig 3B). Paper sampling had a global distribution (see Fig 4A), and a variety of environment types.

## Text analysis

Before creating the LDA Topic Model, the collected articles were aggregated and parsed into unigrams, resulting in over 700,000 different words used across the articles. LDA topic modeling relies on the text corpus that it creates the model from, so words with relatively low occurrences can add confusion to a model. Therefore, all words that did not have at least 5 occurrences were excluded from the dataset to preserve model integrity; this culling left 75,000 words in the dataset.

## LDA topic model

The words output from the text analysis in turn became input texts into the Gensim program to conduct the LDA Topic Model generation. The script created 8 different models, varying from 2 topics to 9 topics. Multiple models allow for visualization and analysis to determine the optimal parameters and is a common practice in machine learning. We evaluated the coherence and perplexity scores for each model. The metric for coherence was based on the extrinsic University of California Irvine (UCI) metric, which compares every word in the corpus to

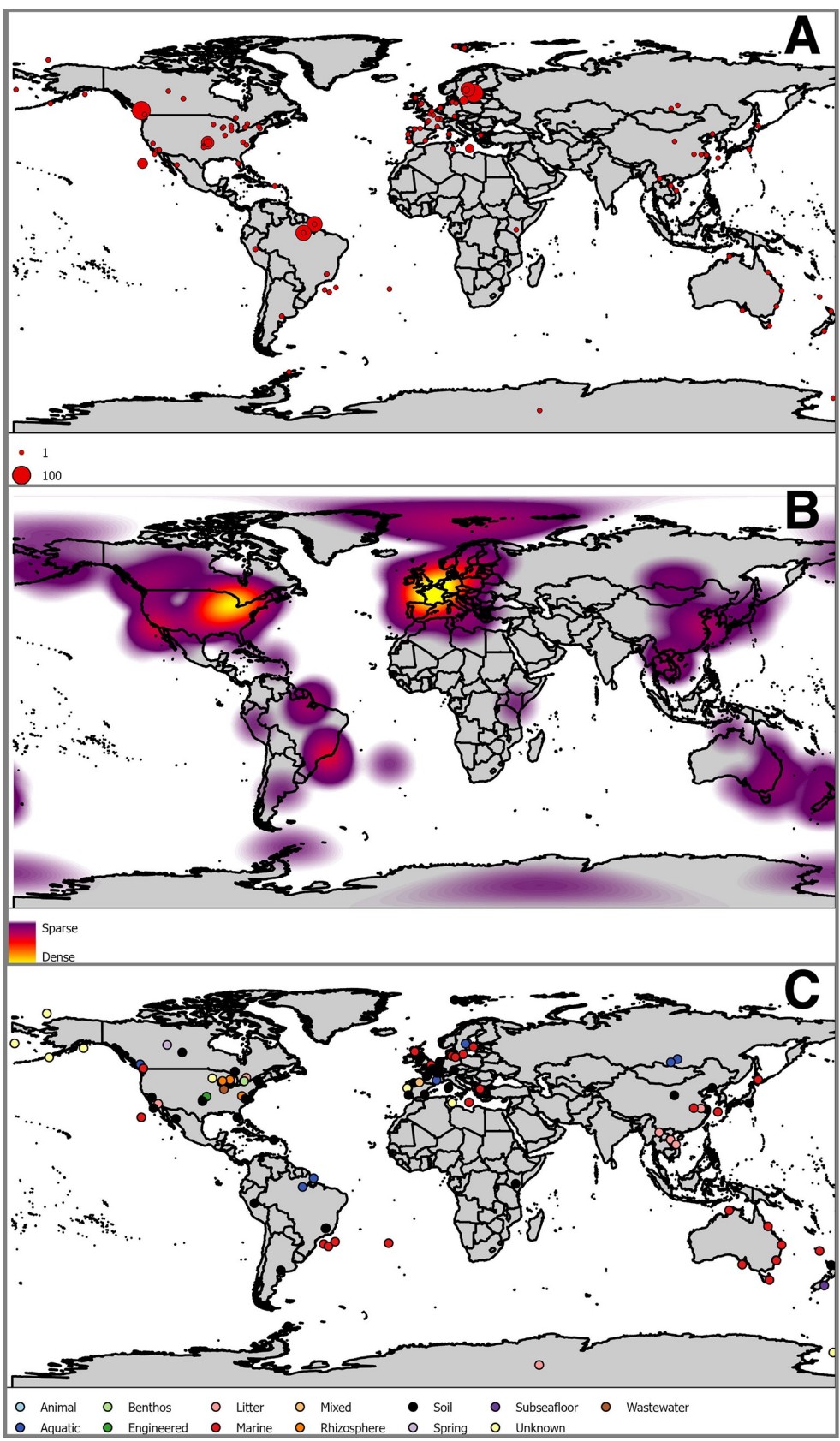

**Fig 4. Sampling locations were derived for each record.** If coordinates for sample sites were provided in the original paper, those were used. If they were not, the centroid of the most precise geographic entity was used. Panel A shows these data with the symbol size scaled to the number of samples indicated at each location. Panel B shows a kernel density of sample locations. Panel C shows the substrate class at each sample locations. Together these locations show a global distribution of *C. neoformans*. It is likely that these locations have sampling bias. Constructed from vector base layers from Natural Earth [45].

every other word in the corpus [46]. By default, the metric Gensim uses for perplexity calculation is the negative logarithmic expansion. Both plots are shown in Fig 5.

Each LDA model created has an associated number of topics, coherence score, and perplexity, which is plotted in Fig 6. The selected LDA model returned three topics and topic probabilities. Each topic had associated words weighted by their relevance to the topic. To determine which number of topics is the optimal amount, another visualization tool, pyLDAvis [47,48] is used. pyLDAvis showed good separation between the topics on the intertopic distance map. The intertopic distances were computed within the algorithm, based on Jensen-Shannon divergence calculations. The centers of the default topic circles are laid out in two dimensions according to a multidimensional scaling (MDS) algorithm that is run on the inter-topic distance matrix. The number of topics can be correlated to the dimensionality of the topic, so MDS reduces dimensionality while maintaining the distance between objects despite being on the same plane with scaling. The higher the separation distance without topic overlap, the better the model is. The model with three topics had the best separation in the group, along with the highest coherence score. Therefore, it is the optimal model and we proceeded to further analyze its parameters.

By default, the Gensim created models have the top words for each topic and their associated weights for each word in comparison to the topic.

In order to relate the topics and top words within each topic to the original documents, we created a script with the aid of the pandas package. Pandas allowed for data table manipulation, and the script enabled us to combine the two data tables into one, and sort the values according to the classification label. The script enabled the visualization of what the LDA model produced: three topic groups with associated words and weights within each group. By examining the words, their associated weights within each topic, and subject matter knowledge on the text corpus, it was possible to determine whether the topic was dominated by either positive papers or negative papers. Topic zero and Topic one share some similarities in words and environments, while Topic two has a different set entirely. The words in Topic two include

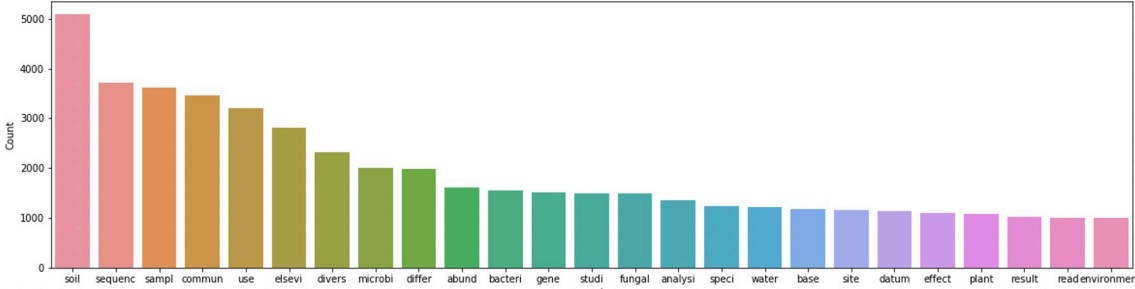

**Fig 5. Count of the most common words across papers positive and negative.** excludes stopwords. Our results show that "crop", "miner", "manag-", "forest-", "litter", "fertil-", "wheat", "contamin-", "amend-", and "root" have the highest association probabilities in the assignment of articles from groups of positive or negative hit papers to topics. The figure is a sample representation of the most popular words found across the text corpus that were statistically relevant after filtering for occurrences. This reveals a potential association between the most common words and environments that contain soil and rhizosphere. Only words above a count of 1000 shown.

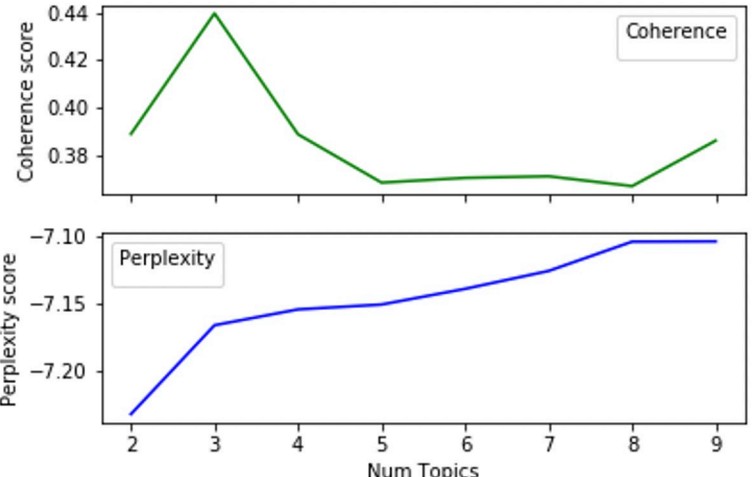

**Fig 6. Graph signifying the coherence score performance in comparison to the number of topics for each model.**
Coherence and Perplexity were calculated at different numbers of topics. Coherence is highest at 3, which indicates
that three topics should be used in the model.

crop, miner-, forest, root, fertil-; these words have strong associations to soil and woody envi-
ronments, while the other topics have associations with warm, watery environments.

Two of the keywords out of the negative topic are "viral" and "ocean" which seems to indicate
that there may be differences in subject matter in these articles (see Fig 7, Topic 1). The middling
topic (Fig 7, Topic: 0) has words like "protist," "manure," and "edna," and "fish." Lastly, the posi-
tive papers (Fig 7, Topic: 2) have words like, "root," "litter," "forest," and "management." Positive
paper words like "root," "litter," "forest," should be describing soils and rhizospheres; Papers
which primarily study "soil" are the largest share of positive hit papers (see Fig 3).

## Random forest evaluation

There is stochasticity in the results from the random forest classifier vary from each iteration
of the script; the results can deviate based on how the decision trees are created and the

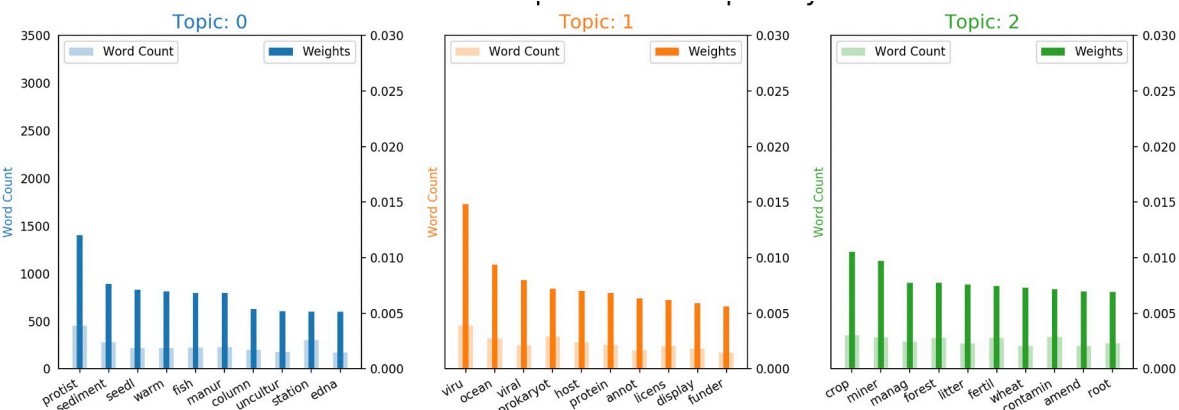

**Fig 7. shows the word count and word weight for the top lemmatized (truncated) words in each topic based on their attributes and by
manipulating the outputs of the matplotlib library.** The most common words in each topic can give insights to what the potential topics are.
They need to be extrapolated by the researchers with subject matter experts based on the text corpus given to the model and words present in
the topics.

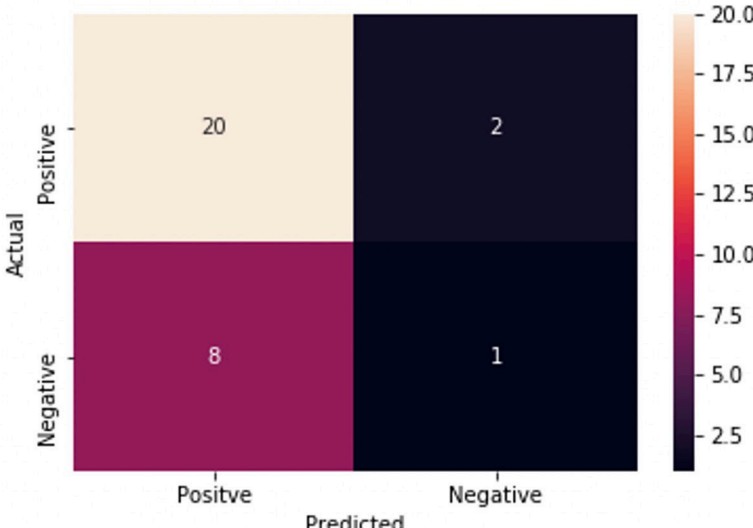

**Fig 8. Confusion matrix from random forest classifier with heatmap indicator for added visualization.** Starting from the top left corner, this square indicates the number of true positives. Moving clockwise, the next square is the false negative area, with the true negative square below it. The last square in the bottom left corner refers to the false positives.

majority voting process. By using the default settings from Sklearn's random forest classifier package, a confusion matrix and AUC-ROC curve were created from the best iteration of the script from 50 trials, see Figs 8 and 9 and Table 2.

## Discussion

Our results show that there is a link between C. neoformans and wood decomposition confirming that this fungus lives in the environment as saprophyte with a preference for wood

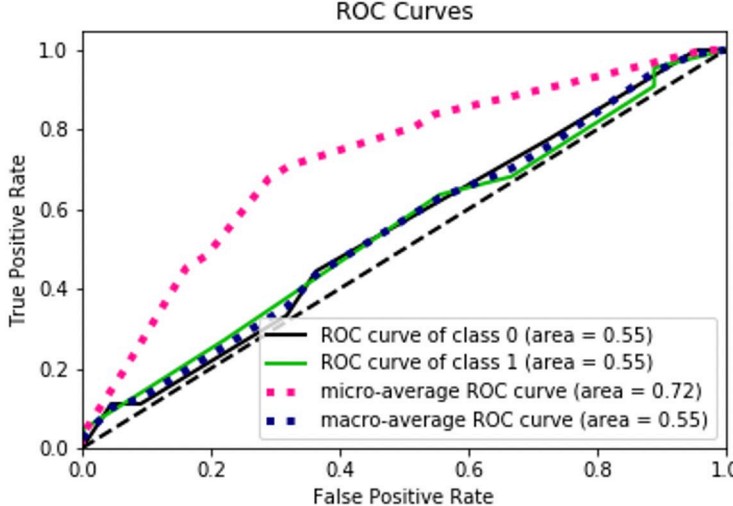

**Fig 9. Graph of the Area Under the Curve—Receiver Operating Characteristic curve, generated by Scikit-plot.** By default, the curve algorithm splits the ROC Curve into three parts. The ROC curve by the classes, the micro-average of the ROC Curve, and the macro-average of the ROC Curve. The micro-average shows that the model predicts better than a random assignment of papers to models.

**Table 2. Summary table of accuracy, precision, recall, f1-score, and AUC, and ROC Score from the random forest classifier.**

| Accuracy | Precision | Recall | F1 Score | AUC Score | ROC Score |
|----------|-----------|--------|----------|-----------|-----------|
| 0.67 | 0.71 | 0.90 | 0.80 | 0.77 | 0.55 |

substrates. This characteristic makes C. neoformans able to occupy a multitude of environmental niches worldwide. These conclusions confirm the validity of the methodology here applied. Our *de novo* topic generation fell into three topics, a topic consisting of positively labeled articles, a middling topic, with both negatively labeled and positively labeled articles, and a topic, consisting of negatively labeled papers, indicating that environmental keywords makes them more similar to each other than to the randomly selected negative topics. Since LDA was used in topic modeling, a true unsupervised algorithm, only extrapolations of the topics can be made, the are essentially bags of words with paper associations. However, the ability to simultaneously characterize hundreds of journal articles, and find the statistically relevant topics and associated words, is a powerful tool in general information gathering and identification of hidden/inferable connections between groups of topics. With the increased use and reduced costs of high-throughput sequencing technology to sample biological diversity in the environment, the number of potentially vastly informative—yet not fully explored—datasets are likely to greatly increase. Data-mining this increasingly large corpus will become crucial to make the most efficient use of these data.

The accuracy, precision, and recall scores for our results are above an average, no-skill model, and are consistent across various iterations from the classifier. In addition, the high F1 Score indicates that the classifier is accurately classifying the values as their correct type, which is ideal for any classification method. In the AUC-ROC curve, the scores revealed interesting aspects about classification performance. The macro ROC score was heavily affected by the per-class ROC scores, while the micro-average was substantially greater. The Micro-average ROC curve is the weighted average ROC of both classes, which is the most informative due to the class imbalance, indicating that the macro average's, or the overall average, was decreased by the presence of false positives and false negatives. However, the micro average is appropriate since we want to classify positively identified papers that have *C. neoformans*, so having a higher weight for correctly classifying positive papers is ideal. Our accuracy in correctly classifying positively labeled papers was reinforced with the resulting AUC score, which can also be interpreted as the probability that a positive instance of a classification is ranked higher than a negative instance of a classification. The similarity between the AUC score and the micro-average ROC score displays similar results, which is that the classifier is correctly classifying positively identified papers at a moderate level (See Table 2).

From a technical perspective, the relatively small number of positive hit papers proved to be challenging for a variety of reasons, including the skew of the extant articles towards positive results. In the data cleaning phase, there was a 90% reduction in the total amount of words was due to the thresholds we set for higher model performance. This reduction would account for single use words, or words shared by only a few articles. Without a high enough count of words and documents, evaluating a model becomes ineffective. If there were a bigger corpus of papers, the output of the initial NLP would have had a higher count of words and word frequencies, so the overall words list(s) to run the LDA against would have been more sizeable, providing more data points for the LDA model to utilize. In addition, the class imbalance between the "positive" and "negative" labeled papers was not ideal compared to a more equal distribution which would have allowed a more standard analysis and approach in the classifier evaluation. While, for the most part, these challenges were circumvented in our approach and

pipeline, utilizing even more computational resources may result in better model prediction. LDA is dependent on datasets size and text corpus content to give the best dirichlet distribution. The number of topics prior to beginning of the LDA analysis may lead future researchers to investigate the possibility of Hierarchical Dirichlet Allocation (HDA) which is similar to LDA but is independent of an initial topic number [49]. In addition, using neural network techniques, like the lda2vec framework based on the word2vec neural network framework [50], may lead to even more significant results due to advantages deep neural networks have over standard classification methods [42,51]. Similarly, with standardization of the metadata content in metabarcoding studies, work like ours would not need to rely as much on Natural Language processing. More traditional meta-analysis methods like non-linear statistical models could be utilized to further determine not only more about the range of environmental features where *C. neoformans* is found, but in all likelihood expand out estimation of its range. There are some limitations caused by the data itself, these limitations included the massive reduction in words from the documents to words that made it into the text corpus for analysis in genism; If even more papers were used then there would be more robust results/analyses. In light of the data limitations approached by this paper we advocate for more metadata standardization (e.g. similar units, required measurements for certain kinds of studies) and more standardization of reporting in journal articles (i.e. required detail in methodology). These small steps can save time in meta-analysis and possibly help NLP.

We analyzed *C. neoformans*, an important human pathogen which is known to have a broad distribution in nature and a few recognized ecological niches, to identify additional potential cryptic niches. It is important to note that the same approach could be used to track different organisms, such as more geographically localized disease agents. Furthermore, the same approach can also be extended to bacterial barcode sequences (such as the 16S rDNA gene region). Given a large enough corpus of relevant articles, this technique could be used to track the ecological niche and geographical location of specific disease agents, an important aspect of biosurveillance. In this work on *C. neoformans* we reinforce the idea of a global distribution, however, the niche of *C. neoformans* is still not well known, which raises the question, if the pathogen has a global distribution, why does it disproportionally affect some populations more than others? And could how *C. neoformans* interacts with its environment be part of that. Importantly, as environmental metagenomic datasets are sampled often organisms that are not the focus of the metagenetic paper will be found, but not reported, as in the case of the corpus used in this work, *C. neoformans* was found, but no reference to *C. neoformans* was made. This analysis sidesteps that problem.

## Acknowledgments

The authors would like to thank The Notre Dame Center for Research computing for supporting the computational infrastructure that this analysis this work was run on, as well as Daniel A. Molik for providing software development computing hardware consultation. The authors also thank Emmet Flynn and Paul Brunts whose undergraduate data science course work inspired some of the machine learning aspects of this paper. Finally, the authors thank the peacefulness and quiet solemnity of the Delbruck Building's back patio for providing the scene of most of this article's ideas.

## Author Contributions

**Conceptualization:** David C. Molik, Benjamin Roche.

**Data curation:** David C. Molik, DeAndre Tomlinson, Shane Davitt, Natalie Meyers.

**Formal analysis:** David C. Molik, DeAndre Tomlinson, Shane Davitt.

**Investigation:** David C. Molik, DeAndre Tomlinson, Shane Davitt.

**Methodology:** David C. Molik, DeAndre Tomlinson, Eric L. Morgan, Matthew Sisk, Benjamin Roche.

**Project administration:** David C. Molik.

**Resources:** Natalie Meyers, Michael E. Pfrender.

**Software:** David C. Molik, DeAndre Tomlinson.

**Supervision:** David C. Molik, Natalie Meyers, Michael E. Pfrender.

**Validation:** David C. Molik, DeAndre Tomlinson.

**Visualization:** David C. Molik, DeAndre Tomlinson, Shane Davitt, Matthew Sisk.

**Writing – original draft:** David C. Molik, DeAndre Tomlinson, Shane Davitt, Natalie Meyers.

**Writing – review & editing:** David C. Molik, DeAndre Tomlinson, Shane Davitt, Eric L. Morgan, Matthew Sisk, Benjamin Roche, Natalie Meyers, Michael E. Pfrender.

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
