## [Decision Letter · Decision Letter 0]

5 Nov 2020

Dear Mr Molik,

Thank you very much for submitting your manuscript "Combining Natural Language Processing and Metabarcoding to Reveal Pathogen-Environment Associations" for consideration at PLOS Neglected Tropical Diseases. As with all papers reviewed by the journal, your manuscript was reviewed by members of the editorial board and by several independent reviewers. In light of the reviews (below this email), we would like to invite the resubmission of a significantly-revised version that takes into account the reviewers' comments. 

We cannot make any decision about publication until we have seen the revised manuscript and your response to the reviewers' comments. Your revised manuscript is also likely to be sent to reviewers for further evaluation.

Sincerely,

Peter John Myler, Ph.D.

Associate Editor

Todd Reynolds

Deputy Editor

Reviewer's Responses to Questions

**Key Review Criteria Required for Acceptance?**

**Methods**

-Are the objectives of the study clearly articulated with a clear testable hypothesis stated?

-Is the study design appropriate to address the stated objectives?

-Is the population clearly described and appropriate for the hypothesis being tested?

-Is the sample size sufficient to ensure adequate power to address the hypothesis being tested?

-Were correct statistical analysis used to support conclusions?

-Are there concerns about ethical or regulatory requirements being met?

Reviewer #1: Methods applied in this study are appropriate but the limited number of sample analyzed could drive to wrong conclusions.

The first paragraph concerning methods is too long and sometimes difficult to understand.

I suggest to move the background about validation of the computer methodologies in the introduction motivating the choice of them. 

As introduction of methods I suggest to list briefly the different steps of the analysis as reported in Fig. 1 which will be then deeply described separately in the following paragraphs.

Line 214. Change "were positive for C. neoformans" to "were specific for C. neoformans".

Line 214. Add accession number of the sequences used as queries.

Figure 2. Symbols in the legend have been converted in squares so it is not possible to follow the explanation of the diagram.

 Line 290. Abbreviation PCA should be explained at the first citation.

Textbox 1, 2 and 3 should be removed since they add a lot of confusing information.

I suggest just to report in the methods the concept and calculation of precision, recall, accuracy, F1, false positive and negative, and true positive and positive.

Reviewer #2: There is a significant amount of detail about various aspects of the method and lengthy explanations provided in text boxes. The methods seem appropriate, but the question being asked is somewhat vague and the application of the various methods is not always clear. It was not clear how the txt analysis was culled from 75,000 words down to the 25 presented in Figure 4. The authors chose Gensim python program to conduct the LDA topic model generation, but the authors do not state why this program was chosen; if there are alternatives or if it is considered the standard method for this type of analysis.

Reviewer #3: No new analysis required.

**Results**

-Does the analysis presented match the analysis plan?

-Are the results clearly and completely presented?

-Are the figures (Tables, Images) of sufficient quality for clarity?

Reviewer #1: Figure 4. Have different colors of bars any meaning?

Line 387. Gensim software should be first cited in the methods with a reference.

Lines 417-424. Topic 1 is not described.

Random Forest Evaluation. In this paragraph results should be reported in detail.

Figure 7. Which is the unit of the bar on the right-side of the picture?

Figure 8. I suppose class 0 refer o negative papers and class 1 positive papers but this should be better explained.

Table 2. From your definitions precision is the proportion of true positives out of all the others. Therefore in this case 20/31= 0.64. Why you reported 0.71? If my calculation is right then also F1 score should be 0.77 (0.64+0.9 /2).

Reviewer #2: It is not clear what topics were used to model, as the authors list 25 potential topic words excluding stop words. They state that the actual model apparently had three topics, but it wasn’t clear which three were included. Or if they modeled each set using an overlapping 3 topic word model. The significance of Figure 6 is not clearly explained. The final model presented in Figure 7 identified 20 true positives out of 31 tested samples, which is only 65%. It is reported in Figure 8 that the micro-average of the two ROC curves is 0.72, but each of those have an area of 0.55, so it is not at all clear how they obtained an average of 0.72. They also do not explain the significance of either micro- or macro-average. The data are presented with very little attempt to explain how they inform the prediction of where C. neoformans might be found or what it means in the context of the question being asked.

Reviewer #3: Additional characterization of the positive hits to include geographic location and/or change over time in geographic location or other information about how the sequencing sample was collected.

**Conclusions**

-Are the conclusions supported by the data presented?

-Are the limitations of analysis clearly described?

-Do the authors discuss how these data can be helpful to advance our understanding of the topic under study?

-Is public health relevance addressed?

Reviewer #1: Lines 449-451. C. neoformans is worldwide distributed, it is not concentrated in Sub-Saharian area!!!!

Reviewer #2: It is difficult to ascertain what conclusions the authors have obtained from this study. They state that it provides support for the previously suggested linkage between C. neoformans and decomposing wood. This is neither a novel nor unexpected result. The link between C. neoformans and decomposing wood is not in doubt as there are dozens of papers that have identified C. neoformans in various environmental locations associated with rotting wood. The application of NLP is a somewhat novel approach, but the utility of this approach is not very apparent in the unconvincing predictive power of the model that they built. It seems that unless the metagenetic data collected includes information about sampling location, the approach in this paper is unlikely to provide any additional insight as to the environmental origins of the sequenced samples.

Reviewer #3: Additional discussion about implications for public health would enhance the paper.

**Editorial and Data Presentation Modifications?**

Reviewer #1: (No Response)

Reviewer #2: None

Reviewer #3: I'm not sure if the presence of text boxes and other insets fits with the style of the journal. The information given in the textboxes may need to be added into the main text of the article, or given in a glossary at the end.

**Summary and General Comments**

Reviewer #1: In general this study is interesting since it try to obtain new information throughout new computer methodologies able to analyze a big amount of complex data.

On the other hand it presents a lot of steps which present a multitude of variables difficult to control and that can compromise the final result of the analysis.

Reviewer #2: This manuscript presents a potentially novel method for identifying the environmental source of metagenetic data in the short-read archive at NCBI. The proposed method uses natural language processing to evaluate the associated manuscripts for information regarding where samples are were taken. The hypothesis is that sampled data may include genetic data from organisms in addition to those that were intended to be samples. In this study, the authors identify C. neoformans genetic material in SRA Biosamples that had not previously been shown to contain C. neoformans data. However, their results provide no additional information about the environmental source of these data other than that they were found in samples associated with decomposing wood, which is a well-established niche for C. neoformans. They do not make a convincing argument that this method will provide additional insight over what has probably already been collected and provided at the time of sample collection.

Reviewer #3: The manuscript “Combining Natural Language Processing and Metabarcoding to Reveal Pathogen-Environment Associations” by Molik et al is presents an interesting methodology which applies a Machine Learning approach to a specific scientific topic of interest, namely, the habitat of C. neoformans. While the paper is extremely well written and the concept is of interest to the field, I think there a few limitations that should be addressed, and if overcome would enhance the reception of the paper. 

 In the discussion, the author implies that the current consensus in the field is that C. neoformans is more abundant in Sub-Saharan Africa (Line 450) which is certainly not the case. It is well understood that while cases of cryptococcosis occur mainly in sub-Saharan Africa, this is due to HIV-AIDS spread and in these areas, not due to additional presence of the yeast. C. neoformans is known to be cosmopolitan, and has been found in the environment all over the world, which is discussed elsewhere in the paper. The overall conclusion of the paper, that C. neoformans is associated with soils and decomposing wood, seems to not add anything that was not previously known. The idea behind the paper and the methodology leading up to this conclusion is very exciting. If the positive hits could be further mined to add additional pieces of data such as where the positive hits were geographically located, and or the change in the geographic location over time it would strengthen the conclusion and add to what is already known. It would be interesting to know this data so that researchers can get more evidence for distribution and potential exposure and latency in individuals. While I think discussion of the technique and the implications for other pathogens is important, it is also important to provide more discussion about the topic selected for this paper. Some questions to consider in the discussion: How would understanding the ecological niche benefit the Cryptococcus field, what else could be revealed by future analyses with a larger amount of papers going forward, could this method also be applied to microbiome sequencing data to ascertain potential age/ area of exposure to C. neoformans? Overall I think the approach has a lot of promise and the authors have done a great job of explaining their method and taking a unique approach to investigating a pathogens origin.

PLOS authors have the option to publish the peer review history of their article (what does this mean?). If published, this will include your full peer review and any attached files.

Reviewer #1: No

Reviewer #2: No

Reviewer #3: No
---

## [Decision Letter · Decision Letter 1]

2 Mar 2021

Dear Mr Molik,

Thank you very much for submitting your manuscript "Combining Natural Language Processing and Metabarcoding to Reveal Pathogen-Environment Associations" for consideration at PLOS Neglected Tropical Diseases. As with all papers reviewed by the journal, your manuscript was reviewed by members of the editorial board and by several independent reviewers. The reviewers appreciated the attention to an important topic. Based on the reviews, we are likely to accept this manuscript for publication, providing that you modify the manuscript according to the review recommendations. 

If you respond appropriately to the comments from Reviewer 1, the manuscript should be suitable for publication.

Sincerely,

Peter John Myler, Ph.D.

Associate Editor

Todd Reynolds

Deputy Editor

If you respond appropriately to the comments from Reviewer 1, the manuscript should be suitable for publication.

Reviewer's Responses to Questions

**Key Review Criteria Required for Acceptance?**

**Methods**

-Are the objectives of the study clearly articulated with a clear testable hypothesis stated?

-Is the study design appropriate to address the stated objectives?

-Is the population clearly described and appropriate for the hypothesis being tested?

-Is the sample size sufficient to ensure adequate power to address the hypothesis being tested?

-Were correct statistical analysis used to support conclusions?

-Are there concerns about ethical or regulatory requirements being met?

Reviewer #1: Authors replied sufficiently to the comments

Reviewer #2: The methods are better explained compared to the first submission and meet the stated criteria.

Reviewer #3: The methods are articulated and apply to a testable hypothesis. The authors have taken steps to make the description of the method clear and address potential shortcomings in terms of sample size limitations and choice of model.

**Results**

-Does the analysis presented match the analysis plan?

-Are the results clearly and completely presented?

-Are the figures (Tables, Images) of sufficient quality for clarity?

Reviewer #1: Authors replied sufficiently to the comments

Reviewer #2: Yes, the results meet all of the stated criteria.

Reviewer #3: Authors have addressed the lack of geographic analysis by adding an additional figure and addressing it in the text as well.

**Conclusions**

-Are the conclusions supported by the data presented?

-Are the limitations of analysis clearly described?

-Do the authors discuss how these data can be helpful to advance our understanding of the topic under study?

-Is public health relevance addressed?

Reviewer #1: see general comments

Reviewer #2: The edits to the conclusions from the first submission make a clearer case for how this approach could be used to study other underreported pathogens and also why their chosen approach is a better method than neural language processing.

Reviewer #3: The authors have expanded their discussion as suggested by the reviewers and fixed a potentially misleading line about geographic distribution.

**Editorial and Data Presentation Modifications?**

Reviewer #1: no comments

Reviewer #2: None

Reviewer #3: None

**Summary and General Comments**

Reviewer #1: The authors replied correctly to most of the comments but one point concerning conclusions still remains to be revised.

Sentence from line 567 to line 570 should be deeply modified as follows:

“Our results show that there is a link between C. neoformans and wood decomposition confirming that this fungus lives in the environment as saprophyte with a preference for wood substrates. This characteristic make C. neoformans able to occupy a multitude of environmental niches worldwide. These conclusions confirm the validity of the methodology here applied. Our de novo…..”

Statement about findings of C. neoformans outside Sub-Sahara is not correct and should be deleted.

Figure legends are lacking.

Reviewer #2: The authors responded well to the critiques raised by the reviewers and I feel the manuscript is suitable for publication.

Reviewer #3: The authors have taken appropriate measures to address the concerns of all three reviewers and have added an additional figure and amended the text to clarify confusing language and make the data more accessible. Although the novelty and significance was initially called into question, the updated manuscript emphasizes the novelty of the technique and the new figure adds information about C. neoformans location that should be of interest to the field.

PLOS authors have the option to publish the peer review history of their article (what does this mean?). If published, this will include your full peer review and any attached files.

Reviewer #1: No

Reviewer #2: No

Reviewer #3: No

Figure Files:

Data Requirements:

Reproducibility:

References

---

## [Editor Report · Decision Letter 2]

9 Mar 2021

Dear Mr Molik,

We are pleased to inform you that your manuscript 'Combining Natural Language Processing and Metabarcoding to Reveal Pathogen-Environment Associations' has been provisionally accepted for publication in PLOS Neglected Tropical Diseases.

Best regards,

Peter J Myler, Ph.D.

Associate Editor

Todd Reynolds

Deputy Editor

Thanks for addressing the remaining issues raised by Reviewer 1.

---

## [Editor Report · Acceptance letter]

1 Apr 2021

Dear Mr Molik,

We are delighted to inform you that your manuscript, "Combining Natural Language Processing and Metabarcoding to Reveal Pathogen-Environment Associations," has been formally accepted for publication in PLOS Neglected Tropical Diseases.

Best regards,

Shaden Kamhawi

co-Editor-in-Chief

Paul Brindley

co-Editor-in-Chief
